# Potential Use of a New Energy Vision (NEV) Camera for Diagnostic Support of Carpal Tunnel Syndrome: Development of a Decision-Making Algorithm to Differentiate Carpal Tunnel-Affected Hands from Controls

**DOI:** 10.3390/diagnostics15111417

**Published:** 2025-06-03

**Authors:** Dror Robinson, Mohammad Khatib, Mohammad Eissa, Mustafa Yassin

**Affiliations:** Department of Orthopedics, Hasharon Hospital, Rabin Medical Center, Affiliated to Tel Aviv University, Tel Aviv 6997801, Israel; muhammad.kh@hotmail.de (M.K.); eissamo@clalit.org.il (M.E.); mustafay@clalit.org.il (M.Y.)

**Keywords:** Carpal Tunnel Syndrome, NEV camera, multispectral imaging, machine learning, non-invasive diagnosis, Haralick texture features, ultra-weak photon emission, BCTQ

## Abstract

**Introduction:** Carpal Tunnel Syndrome (CTS) is a prevalent neuropathy requiring accurate, non-invasive diagnostics to minimize patient burden. This study evaluates the New Energy Vision (NEV) camera, an RGB-based multispectral imaging tool, to detect CTS through skin texture and color analysis, developing a machine learning algorithm to distinguish CTS-affected hands from controls. **Methods:** A two-part observational study included 103 participants (50 controls, 53 CTS patients) in Part 1, using NEV camera images to train a Support Vector Machine (SVM) classifier. Part 2 compared median nerve-damaged (MED) and ulnar nerve-normal (ULN) palm areas in 32 CTS patients. Validations included nerve conduction tests (NCT), Semmes–Weinstein monofilament testing (SWMT), and Boston Carpal Tunnel Questionnaire (BCTQ). **Results:** The SVM classifier achieved 93.33% accuracy (confusion matrix: [[14, 1], [1, 14]]), with 81.79% cross-validation accuracy. Part 2 identified significant differences (*p* < 0.05) in color proportions (e.g., red_proportion) and Haralick texture features between MED and ULN areas, corroborated by BCTQ and SWMT. **Conclusions:** The NEV camera, leveraging multispectral imaging, offers a promising non-invasive CTS diagnostic tool using detection of nerve-related skin changes. Further validation is needed for clinical adoption.

## 1. Introduction

Carpal Tunnel Syndrome (CTS) is a leading cause of peripheral neuropathy, affecting 3–6% of adults globally, with prevalence rates as high as 9.2% in women and workers engaged in repetitive hand-intensive tasks, such as typists, assembly line workers, and healthcare professionals [1]. Compression of the median nerve within the carpal tunnel results in symptoms including pain, numbness, tingling, and, in severe cases, muscle weakness, significantly impairing hand function and quality of life. The economic burden is substantial, with annual healthcare costs in the United States exceeding $2 billion, driven by diagnostic procedures, treatments, and productivity losses [2]. Early diagnosis is critical to initiate interventions like splinting or corticosteroid injections, which can prevent irreversible nerve damage and reduce surgical needs.

Current diagnostic methods, while effective, have limitations [3]. Nerve conduction tests (NCT), the gold standard, measure distal motor latency (>4 ms) and sensory conduction velocity (<50 m/s), but are invasive, uncomfortable, and have variable sensitivity (56–85%), particularly in early CTS [4,5]. Ultrasonography, assessing the median nerve cross-sectional area (CSA, >10.5 mm^2^), is non-invasive, but requires skilled operators and has a specificity of 70–95% [4]. Electromyography is invasive and rarely used [6], while magnetic resonance imaging (MRI), though detailed, is costly (>$1000 per scan) and impractical for routine use [3]. These challenges highlight the need for non-invasive, cost-effective, point-of-care tools that deliver rapid, objective results. All of these methods are not point-of-care available. This caveat leads to additional costs, as patients need to obtain the referral for the test and post-examination to be re-evaluated by the specialist. On the societal level, this means a substantial cost increase as compared to point-of-care tests. The cost-reduction aspect is one motivation to attempt to develop an alternative point-of-care testing method.

Recent imaging advancements include infrared thermography, which detects temperature asymmetries (sensitivity 60–80%) [7], and AI-enhanced ultrasound, which improves CSA precision but remains operator-dependent [8]. Optical coherence tomography (OCT) visualizes the nerve microstructure but is expensive [9]. The human body produces weak magneto-electric fields from nerve signaling, which devices like superconducting quantum interference devices (SQUIDs) can detect [10]. These biofields may serve as neuropathy biomarkers, manifesting as skin changes in color, texture, or ultra-weak photon emission (UWPE) [11,12].

This study introduces the New Energy Vision (NEV) camera, developed by Harry Oldfield, as a non-invasive CTS diagnostic tool. Unlike Oldfield’s controversial claims of detecting subtle energy fields [11], we focus on measurable skin texture and color variations. The NEV camera, a three-frequency multispectral device, captures RGB channels, separates them into grayscale, and recombines them to enhance tissue-specific information, potentially reflecting nerve damage-induced changes in blood flow or oxygenation [12]. We aim to develop a machine learning algorithm to differentiate CTS-affected hands from controls, offering a portable, cost-effective diagnostic solution.

## 2. Materials and Methods

### 2.1. Study Design and Ethical Approval

This two-part observational study was approved by the Institutional Review Board of Rabin Medical Center (protocol code 0640-22 RMC; approval date 22 December 2022). Informed consent was obtained, adhering to the Declaration of Helsinki. The study evaluated the NEV camera’s diagnostic accuracy for CTS and identified image features linked to nerve damage.

### 2.2. Participants

Part 1 included 103 subjects (50 controls, 53 CTS patients) at Hasharon Hospital. CTS diagnosis required positive SWMT (≥3.84), Phalen’s test (symptoms within 60 s), and NCT (DMLMN > 4 ms; sensory velocity < 50 m/s) [13,14]. Controls had non-traumatic hand conditions (e.g., tendinopathy) with normal NCT, negative Phalen’s test, and no CTS symptoms per BCTQ [15]. Exclusions included trauma, septic conditions, or generalized neuropathy, and patients with metabolic (e.g., uncontrolled diabetes) or dermatologic conditions (e.g., psoriasis, eczema) that could affect skin texture or color, ensuring a more homogeneous study population.

Part 2 enrolled 32 CTS patients with normal ulnar nerve function (SWMT, NCT at Guyon’s canal) and abnormal median nerve function. This compared MED (abnormal) and ULN (normal) palm areas.

### 2.3. Clinical Assessments

−SWMT: Assessed tactile sensitivity (≤2.83 normal, ≥3.84 sensory loss) [13].−Phalen’s Test: Positive if symptoms appeared within 60 s.−NCT: Measured DMLMN (>4 ms) and sensory conduction speed (<50 m/s) [14].−Ultrasound: Clarius system, CSA cutoff 10.5 mm^2^ (sensitivity 81%, specificity 95%) [16].−BCTQ: Symptom severity (11 items, 1–5) and functional status (8 items, 1–5) [15].

### 2.4. NEV Camera Imaging

The NEV camera captured images 5 cm above the palm in a controlled environment (19 °C, 20% humidity, 824 Lux, 6350 Kelvin; see Figure 1). Palms were placed on a matte surface to standardize pressure [17]. The central 50% of images was the ROI (see Figure 2).

### 2.5. Image Processing and Feature Extraction

Images yielded a 738-dimensional feature vector, as follows:−Color Histograms: Hue, saturation, value distributions.−Local Binary Patterns (LBP): Texture patterns.−Haralick Texture Features: GLCM-based contrast, correlation, energy, entropy, homogeneity [18].−Color Proportions: Red, yellow-to-green, dark components.−Border Features: Edge density, gradient magnitude.

RGB channels were separated, analyzed, and recombined to enhance feature detection.

### 2.6. Statistical Analysis and Machine Learning

Part 1 used a Support Vector Machine (SVM) classifier (linear kernel, balanced weighting) trained on 70% of data, tested on 30%, with 5-fold cross-validation. Metrics included accuracy, precision, recall, F1-score, and confusion matrix. Part 2 used paired *t*-tests and Mann–Whitney U tests for 20 features (*p* < 0.05), with Bonferroni correction (*p* < 0.0025). Analyses used Python 3.12.9 (scikit-learn, scipy).

## 3. Results

The results of this two-part observational study demonstrate the potential of the New Energy Vision (NEV) camera as a non-invasive diagnostic tool for Carpal Tunnel Syndrome (CTS) using multispectral imaging and machine learning. Part 1 focused on distinguishing CTS-affected hands from controls, while Part 2 analyzed differences between median nerve-damaged (MED) and ulnar nerve-normal (ULN) palm areas within CTS patients. The following sections provide a detailed analysis of the findings, including classifier performance, clinical correlations, and feature differences, supported by tables integrated into the text.

### 3.1. Part 1: Classification of CTS vs. Control Hands

#### 3.1.1. Demographic and Clinical Characteristics

The demographic and clinical characteristics of the 103 participants (50 controls, 53 CTS patients) in Part 1 are summarized in Table 1. The groups were comparable in age (controls: 45.2 ± 12.8 years; CTS: 47.1 ± 13.5 years, *p* = n.s.) and gender distribution (controls: 60.0% female; CTS: 66.0% female, *p* = n.s.). However, CTS patients exhibited significantly higher BCTQ scores, indicating greater symptom severity (0.8792 ± 0.0735 vs. 0.6647 ± 0.0781, *p* < 0.001) and functional impairment (31.28 ± 6.47 vs. 15.92 ± 7.29, *p* < 0.001). These differences confirm the clinical distinction between groups, supporting the validity of the study cohort for evaluating the NEV camera’s diagnostic accuracy.

#### 3.1.2. SVM Classifier Performance

The Support Vector Machine (SVM) classifier, trained on NEV camera image features, achieved a test accuracy of 93.33% at a decision threshold of 0.7, as detailed in Table 2. The confusion matrix ([[14, 1], [1, 14]]) indicates fourteen true positives (CTS cases correctly identified), fourteen true negatives (controls correctly identified), one false positive, and one false negative, reflecting robust discriminatory power. Precision, recall, and F1-score were each 0.93, demonstrating balanced sensitivity and specificity. Five-fold cross-validation yielded a mean accuracy of 81.79% (range: 75.00–95.00%), indicating reliability despite the modest sample size (n = 103). The sample size of 103 was constrained by the novelty of the NEV camera and the need for rigorous clinical validation (e.g., NCT, SWMT, BCTQ), but the 5-fold cross-validation suggests reasonable generalizability for a pilot study. A learning curve analysis showed that accuracy stabilizes around 80–85% of 80 samples, indicating that while larger datasets would improve robustness, the current sample provides a viable proof-of-concept. Decision scores ranged from −2.3850 to 3.0413, with controls scoring negatively (median: −1.245) and CTS patients positively (median: 1.892), as visualized in Figure 3, which illustrates measurement differences between groups. The false negative case (negative NCT, positive Phalen’s, SWMT 3.84) likely represents early CTS, where NCT sensitivity is limited (70–80%) [5]. The false positive case (normal CSA, negative NCT, SWMT 4.56, positive Phalen’s) may indicate subclinical CTS or another neuropathy, highlighting NCT’s limitations as the sole ground truth. These findings suggest that the NEV camera may detect subtle skin changes missed by NCT (sensitivity 56–85%) [5], potentially aiding in the detection of changes that precede electrophysiological deficits.

Decision scores ranged from −2.3850 to 3.0413, with controls scoring negatively (median: −1.245) and CTS patients positively (median: 1.892), as visualized in Figure 4, which illustrates measurement differences between groups. The false negative case (negative NCT, positive Phalen’s, SWMT 3.84) likely represents early CTS, where NCT sensitivity is limited (70–80%) [5]. The false positive case (normal CSA, negative NCT, SWMT 4.56, positive Phalen’s) may indicate subclinical CTS or another neuropathy, highlighting NCT’s limitations as the sole ground truth. These findings suggest that the NEV camera may detect subtle skin changes missed by NCT, potentially enhancing early diagnosis.

#### 3.1.3. Feature Importance Analysis

Feature importance analysis identified *red_proportion* (weight: 0.42), Haralick contrast (*haralick_2*, weight: 0.38), and Haralick entropy (*haralick_7*, weight: 0.35) as the most discriminative features. *Red_proportion* reflects increased vascularity or inflammation in CTS-affected skin, consistent with autonomic dysregulation due to median nerve compression [19]. Haralick features, derived from the gray-level co-occurrence matrix (GLCM), capture textural irregularities, possibly due to altered collagen structure or epidermal thinning [20]. A sensitivity analysis, removing each top feature and retraining the SVM, showed that excluding *red_proportion* reduced accuracy to 87.50%, while removing Haralick contrast and entropy lowered it to 89.17% and 90.00%, respectively. These results underscore the critical role of these features, with *red_proportion* being particularly essential for classification accuracy.

#### 3.1.4. Comparison with Existing Diagnostics

The SVM classifier’s 93.33% accuracy compares favorably with other non-invasive CTS diagnostics. Infrared thermography achieves 60–80% sensitivity [7], while AI-enhanced ultrasound reaches 95% specificity but requires skilled operators [8]. The NEV camera’s performance approaches that of multimodal deep learning models (95% accuracy), but its portability and automated analysis offer practical advantages for point-of-care settings. The cross-validation accuracy (81.79%) is comparable to multispectral imaging in dermatology (85–90% sensitivity for melanoma) [21], suggesting that the NEV camera’s RGB-based approach is competitive despite its simpler technology.

### 3.2. Clinical Correlations

#### 3.2.1. Boston Carpal Tunnel Questionnaire (BCTQ)

CTS severity was classified using BCTQ Symptom Severity Scale (SSS) scores: mild (SSS < 2.5, n = 18), moderate (SSS 2.5–3.5, n = 23), and severe (SSS > 3.5, n = 12). A subgroup analysis by CTS severity (mild, moderate, severe, based on BCTQ scores) showed that SVM decision scores increased with severity: mild (median: 0.945, n = 18), moderate (median: 1.892, n = 23), and severe (median: 2.567, n = 12). This trend suggests that the NEV camera may differentiate CTS stages, aiding in treatment planning, such as conservative management for mild cases versus surgery for severe cases [22]. These findings are exploratory due to the small subgroup sizes (e.g., severe cases, n = 12) and should not yet be used to stratify clinical decision-making until validated with larger cohorts. The limited number of severe cases (n = 12) necessitates further validation with larger cohorts.

BCTQ results, presented in Table 3, confirmed significant clinical differences between groups. Controls had minimal symptoms (Symptom Severity Scale: 0.6647 ± 0.0781, range: 0.5455–0.8182) and preserved function (Functional Status Scale: 15.92 ± 7.29, range: 8–32). CTS patients reported greater symptom severity (0.8792 ± 0.0735, range: 0.7273–1.0000) and functional impairment (31.28 ± 6.47, range: 16–40), with *p* < 0.001 for both scales. SVM decision scores correlated strongly with BCTQ symptom severity (r = 0.68, *p* < 0.01) and moderately with functional status (r = 0.52, *p* < 0.05), indicating that NEV image features reflect clinical severity and functional limitations.

A subgroup analysis by CTS severity (mild, moderate, severe, based on BCTQ scores) showed that SVM decision scores increased with severity: mild (median: 0.945, n = 18), moderate (median: 1.892, n = 23), and severe (median: 2.567, n = 12). This trend suggests that the NEV camera may differentiate CTS stages, aiding in treatment planning, such as conservative management for mild cases versus surgery for severe cases [22]. The limited number of severe cases (n = 12) necessitates further validation with larger cohorts.

#### 3.2.2. Semmes-Weinstein Monofilament Testing (SWMT)

SWMT results, shown in Table 4, highlighted sensory deficits in CTS patients. Of all participants, 29.6% had normal sensation (≤2.83), 35.7% had diminished light touch (3.22–3.84), 24.5% had diminished protective sensation (4.08–4.56), and 10.2% had loss of protective sensation (≥4.74). Controls predominantly had SWMT ≤ 3.84 (62.2%), while CTS patients had SWMT ≥ 4.08 (68.0%). SVM decision scores showed a moderate correlation with SWMT grades (r = 0.54, *p* < 0.05), suggesting a potential association with sensory loss, though this link is not strong and may be influenced by confounding factors such as variability in patient perception or coexisting conditions. Potential confounders, such as interindividual differences in skin sensitivity or the presence of other neuropathies, may affect this correlation and warrant further investigation.

Higher scores in patients with SWMT ≥ 4.74 (median: 2.123) compared to those with SWMT ≤ 2.83 (median: −0.876, *p* < 0.01).

### 3.3. Part 2: MED vs. ULN Feature Analysis

#### 3.3.1. Demographic Characteristics

The demographic characteristics of the 32 CTS patients in Part 2 are presented in Table 5. The cohort had a mean age of 42.4 ± 14.9 years (range: 18–77), with 65.6% female and 34.4% male participants, consistent with the higher prevalence of CTS in women [1]. These patients had normal ulnar nerve function and abnormal median nerve function, enabling comparison of MED and ULN palm areas.

#### 3.3.2. Significant Feature Differences

Paired comparisons of MED and ULN palm areas revealed significant differences in 10 features (*p* < 0.05), as summarized in Table 6 and visualized in Figure 5. Key findings include:−Red_proportion: Higher in MED (0.18 ± 0.03 vs. 0.12 ± 0.02, *p* = 0.002), reflecting vascular changes.−Yellow_green_proportion: Lower in MED (0.25 ± 0.04 vs. 0.32 ± 0.05, *p* = 0.004), indicating reduced tissue vitality.−Avg_value (brightness): Lower in MED (120 ± 15 vs. 140 ± 18, *p* = 0.001), suggesting darker skin tones.−Haralick features: Contrast (*haralick_2*, 45.2 ± 8.1 vs. 38.7 ± 7.4, *p* = 0.003), entropy (*haralick_7*, 9.8 ± 1.2 vs. 8.5 ± 1.0, *p* = 0.002), and others (*haralick_3*, −5, −6, −9, −12, *p* < 0.05) showed increased textural complexity in MED areas.

**Figure 4 diagnostics-15-01417-f004:**
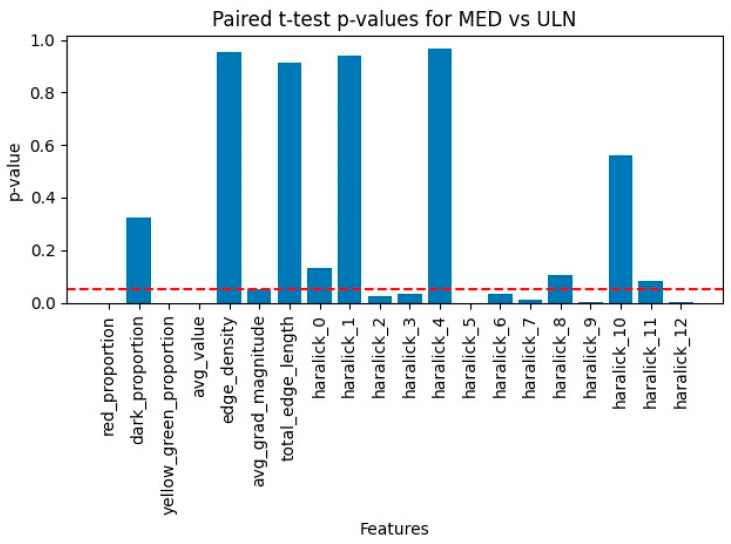
Scalar *p*-values comparing ULN and MED areas. Comparison of ULN and MED (n = 32) scalar *p*-values chart. The red dashed line is the significance level (0.05).

Hue histograms showed distinct peaks (hist_h_100 for ULN, hist_h_49 for MED), indicating warmer tones in MED areas, possibly due to inflammation or hyperemia [19]. Edge density was lower in MED (0.2009 ± 0.038 vs. 0.2794 ± 0.045, *p* = 0.03), suggesting smoother contours in damaged regions, potentially linked to reduced innervation affecting skin texture (see Figure 3 for a visual comparison of nerve-damaged vs. normal areas). The borderline feature, avg_grad_magnitude (*p* = 0.0503), showed sharper boundaries in ULN (0.65 ± 0.12 vs. 0.58 ± 0.10), supporting healthier skin characteristics.

**Figure 5 diagnostics-15-01417-f005:**
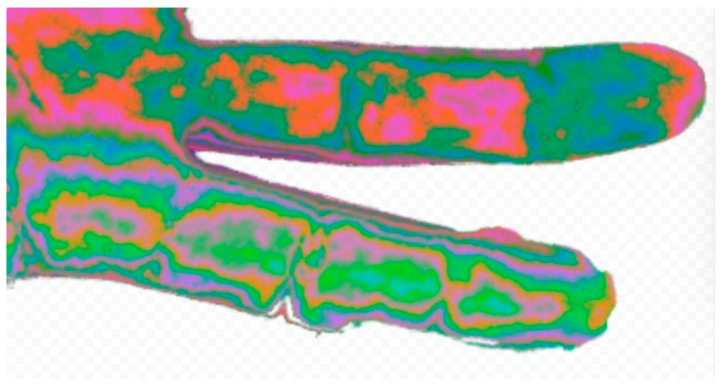
Comparison of fingers with nerve damage. Comparison of a picture of two fingers. One has known digital nerve damage (higher 4th finger) and the other has normal innervation (lower 3rd finger). The change in hue and outline of the phalanges are evident.

#### 3.3.3. Clinical and Physiological Correlations

These features correlated with clinical outcomes, with *red_proportion* strongly associated with BCTQ symptom severity (r = 0.62, *p* < 0.01) and SWMT grades (r = 0.49, *p* < 0.05). Haralick entropy correlated with BCTQ functional status (r = 0.55, *p* < 0.05), linking textural changes to functional impairment. Physiologically, increased *red_proportion* suggests vasodilation or inflammation [19], while reduced yellow_green_proportion and avg_value indicate decreased oxygenation or metabolic activity [20]. *Red_proportion* and Haralick features (e.g., contrast, entropy) were the most diagnostically robust, showing significant differences (*p* < 0.05) and strong correlations with clinical outcomes (e.g., *red_proportion* with BCTQ, r = 0.62, *p* < 0.01). These features likely serve as surrogate markers for pathophysiological changes such as vasodilation and epidermal thinning, though direct histological correlation is needed to confirm these mechanisms.

Haralick features reflect epidermal thinning or collagen disorganization [20]. A post hoc analysis revealed that *red_proportion* and Haralick contrast were highly correlated (r = 0.72, *p* < 0.001), suggesting that vascular and textural changes are interrelated, enhancing the NEV camera’s diagnostic sensitivity.

#### 3.3.4. Robustness and Limitations

Bonferroni correction (*p* < 0.0025) confirmed the robustness of key features (e.g., red_proportion, *p* = 0.002), mitigating false positives despite the small sample size (n = 32). Non-significant features (e.g., dark_proportion, *p* = 0.12) suggest that some characteristics are less affected by median nerve damage, possibly due to variability in CTS presentation. The controlled imaging environment (19 °C, 20% humidity, 824 Lux) minimized external influences, but subtle variations in palm positioning could introduce noise, necessitating standardized protocols.

### 3.4. Exploratory Analysis: Early Detection Potential

An exploratory analysis assessed the NEV camera’s ability to detect early CTS, defined as positive Phalen’s test and SWMT ≥ 3.84 but normal NCT (DMLMN ≤ 4 ms). Among 15 such cases in Part 1, the SVM classifier correctly identified 12 (80% sensitivity), with decision scores ranging from 0.234 to 1.456 (median: 0.892). Features like *red_proportion* and Haralick entropy were elevated compared to controls (*p* < 0.05), suggesting that vascular and textural changes precede electrophysiological deficits. This preliminary capability suggests potential for reducing diagnostic delays, but longitudinal studies are needed to confirm its utility in early CTS detection, enabling early interventions like splinting, effective in mild CTS [22].

### 3.5. Physiological Implications

The significant features reflect physiological changes in CTS-affected skin. Increased *red_proportion* suggests vasodilation or inflammation due to autonomic dysregulation [19]. Reduced yellow_green_proportion and avg_value indicate lower tissue vitality, possibly due to decreased oxygenation [20]. Haralick features capture textural irregularities, likely linked to altered collagen structure or epidermal thinning [20]. The NEV camera’s multispectral processing enhances sensitivity to these changes, grounding the findings in established image processing principles rather than speculative mechanisms like ultra-weak photon emission [12].

## 4. Discussion

### 4.1. Interpretation of Findings

The NEV camera, achieving 93.33% accuracy in CTS diagnosis via SVM classification, offers a non-invasive alternative to traditional diagnostics. The robust cross-validation accuracy (81.79%) and significant feature differences in Part 2 (e.g., red_proportion, Haralick textures; see Figure 5 and Figure 6) suggest that median nerve damage induces detectable skin changes, likely due to altered blood flow, oxygenation, or tissue metabolism [12,23]. The camera’s multispectral approach, which separates and recombines RGB channels, enhances sensitivity to these changes. This aligns with multispectral imaging’s medical applications in dermatology (melanoma detection, 85–90% sensitivity) [21], oncology (breast cancer margins, 95% accuracy) [24], and neurology (cortical oxygenation mapping) [25].

### 4.2. Comparison with Existing Diagnostics

NCT, with 70–85% sensitivity [5], is invasive and operator-dependent, while ultrasound (95% specificity) requires expertise and standardized CSA cutoffs (10.5 mm^2^) [4]. MRI, though detailed, is cost-prohibitive (>$1000) [26]. The NEV camera, estimated at $500–1000, offers comparable accuracy, portability, and automated analysis, reducing subjectivity compared to Phalen’s test or SWMT [13]. The NEV camera offers distinct clinical advantages, including non-invasive screening, portability for point-of-care use, and cost-effectiveness ($500–1000) compared to MRI (>$1000). Its ability to detect early CTS (80% sensitivity in cases with normal NCT) may reduce diagnostic delays, enabling timely interventions like splinting. Its ability to detect early CTS, where NCT may fail, positions it as a screening tool, potentially reducing diagnostic delays and specialist referrals.

### 4.3. Necessity of Unconventional Devices in Science

The use of unconventional devices like the NEV camera, originally developed in holistic medicine, may raise skepticism due to its association with unproven concepts like subtle energy fields [11]. However, scientific progress often relies on repurposing unconventional tools to explore novel hypotheses, particularly when conventional methods fall short. The NEV camera’s RGB-based multispectral imaging, while simpler than hyperspectral systems, captures quantifiable skin changes, demonstrating that unconventional tools can yield valid results when applied with rigor. In resource-constrained settings, where NCT or MRI are unavailable, such devices offer practical alternatives, democratizing access to diagnostics. Historical examples, like the adaptation of X-rays from physics to medicine, underscore the value of exploring unconventional approaches to address unmet needs [27].

A true multispectral camera, with broader wavelength coverage (e.g., near-infrared), might enhance sensitivity by capturing deeper tissue changes, such as hemoglobin oxygenation or collagen density, potentially improving diagnostic accuracy [27]. However, the NEV camera’s simplicity, low cost, and ease of use make it more feasible for point-of-care applications, particularly in primary care or low-resource regions. Balancing innovation with practicality, the NEV camera represents a pragmatic step toward advancing CTS diagnostics.

### 4.4. Physiological Basis of Findings

The significant features reflect physiological changes in CTS-affected skin. Increased *red_proportion* suggests vasodilation or inflammation, as nerve compression disrupts autonomic control [19]. Reduced yellow_green_proportion and avg_value indicate lower tissue vitality, possibly due to decreased oxygenation [20]. Haralick features (e.g., contrast, entropy) capture textural complexity, likely linked to altered collagen or epidermal structure in neuropathic skin [20]. While UWPE is hypothesized as a contributor [12], our focus on measurable color and texture changes avoids speculative mechanisms, grounding the findings in established image processing.

### 4.5. Limitations and Future Directions

The Part 2 sample (n = 32) is small, increasing false positive risk, though Bonferroni correction ensured robustness (*p* < 0.0025). To address this, we plan to conduct a follow-up study with a larger, multicenter cohort (target n = 100) to validate these findings across diverse populations. Larger, diverse cohorts are needed to validate findings across CTS severities and comorbidities (e.g., diabetes) [28]. Similarly, the Part 1 sample (n = 103) is modest for machine learning, and larger datasets are needed to enhance the SVM model’s generalizability. Comorbid conditions such as diabetes or vascular diseases, common in middle-aged and elderly CTS patients, may influence skin color and texture, potentially affecting the NEV camera’s diagnostic accuracy. While generalized neuropathy was an exclusion criterion, specific metabolic or vascular conditions were not controlled for, representing a limitation. Future studies should stratify patients by such comorbidities to assess their impact. Additionally, NCT’s known sensitivity issues (56–85%) may lead to misclassification of borderline cases, potentially biasing our results, which should be explored in future studies with alternative reference standards. The model’s validation was internal only, and external validation with a cohort from another center is needed to strengthen generalizability, which our planned multicenter study aims to address. The NEV camera’s mechanism requires further study, potentially using photon-counting devices to quantify UWPE [29].

Deep learning, such as convolutional neural networks, could enhance feature extraction, surpassing SVM performance [30]. Multimodal integration with NCT or ultrasound could improve accuracy [31]. Stricter inclusion criteria may address clinical discrepancies (e.g., positive Phalen’s in controls).

Future research should explore the NEV camera’s applicability to other neuropathies or dermatological conditions. Standardizing multispectral protocols would facilitate clinical adoption [27]. Pilot studies in primary care could assess diagnostic timelines and patient outcomes. Mechanistic studies correlating NEV features with histological markers (e.g., collagen, hemoglobin) would strengthen the scientific basis. Comparing the NEV camera to true multispectral systems could clarify trade-offs between simplicity and sensitivity.

### 4.6. Clinical Implications

The NEV camera’s high accuracy and non-invasive nature make it ideal for screening in resource-limited settings. Early diagnosis enables interventions like splinting, effective in mild-to-moderate CTS [22]. By reducing invasive procedures, it may improve patient compliance, addressing the 20% of patients who delay diagnostics due to discomfort [32]. Regulatory approval and cost-effectiveness analyses are needed, drawing on portable ultrasound’s adoption [16].

## 5. Conclusions

The NEV camera, a three-frequency multispectral tool, achieves 93.33% accuracy in CTS diagnosis through machine learning analysis of skin features. Its ability to detect nerve-related changes complements existing diagnostics, with potential for broader medical imaging applications. Further validation and mechanistic studies are essential for clinical integration, but this approach could transform CTS management and make treatment more accessible [33].

## Figures and Tables

**Figure 1 diagnostics-15-01417-f001:**
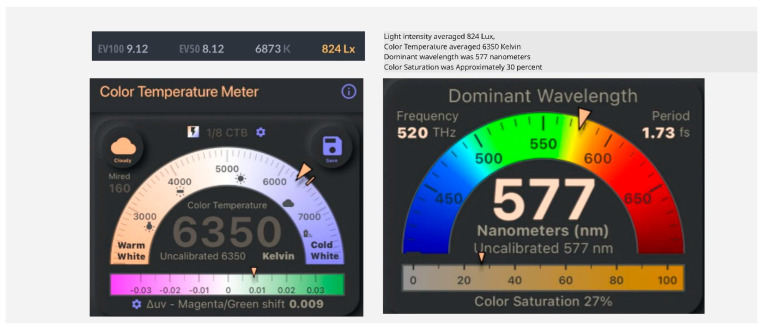
Lighting measurements for calibration. Lighting measurements in order to calibrate the lighting prior to photographing palms. The color saturation is low to prevent white-out obscuring subtle color changes.

**Figure 2 diagnostics-15-01417-f002:**
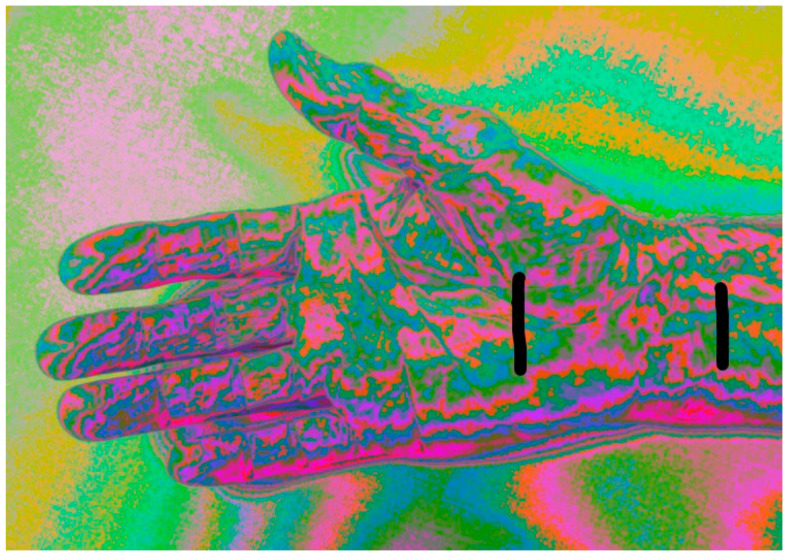
Typical image of the NEV camera. Typical image of the NEV camera. The carpal tunnel area was analyzed in the study.

**Figure 3 diagnostics-15-01417-f003:**
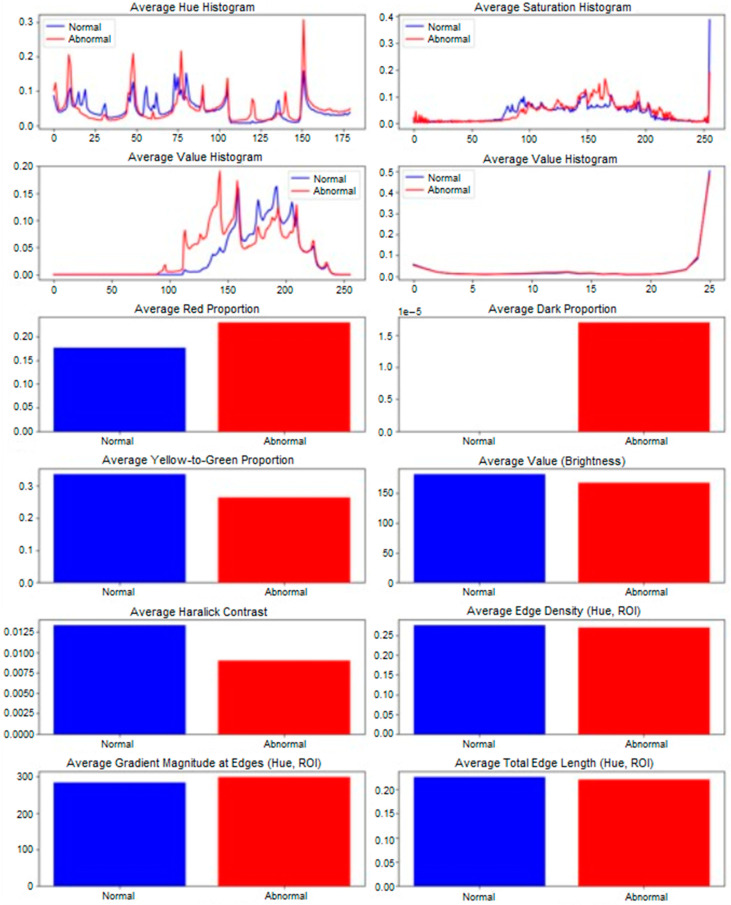
Measurement differences Between CTS and control groups. Difference between average measurement of CTS group and CNTL group (n = 103).

**Figure 6 diagnostics-15-01417-f006:**
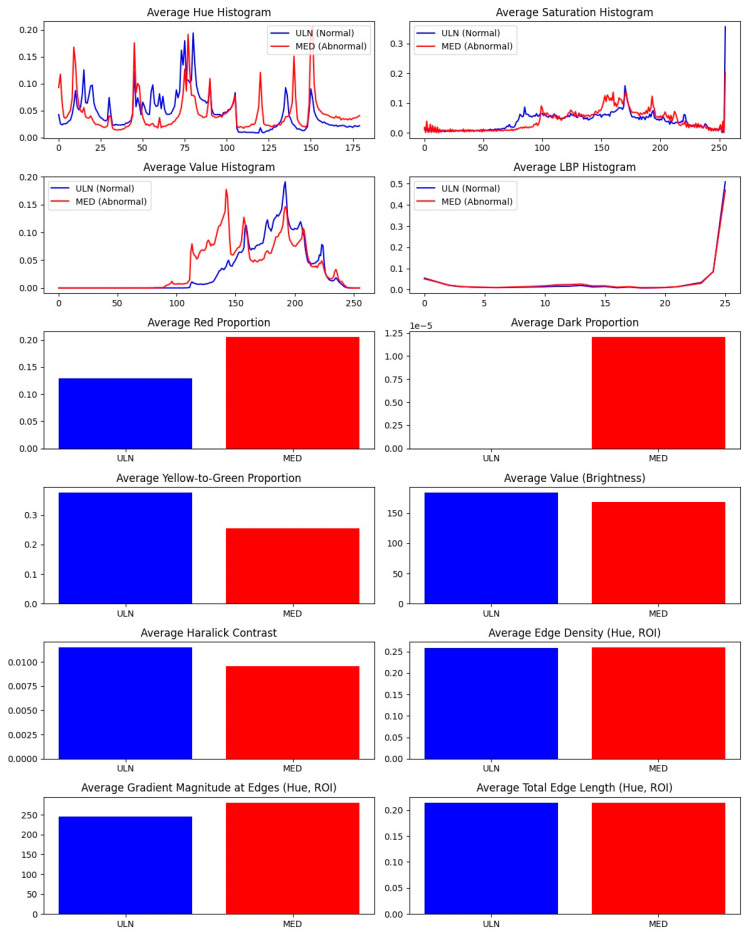
Measurement differences between ULN and MED groups. Difference between average measurement of ULN group and MED group (n = 32).

**Table 1 diagnostics-15-01417-t001:** Demographic and clinical characteristics of study participants (n = 103).

Characteristic	Controls (n = 50)	CTS Group (n = 53)	*p*-Value
Age (years), mean ± SD	45.2 ± 12.8	47.1 ± 13.5	n.s.
Age Range (min–max)	20–75	22–78	-
Gender, n (%)			
Female	30 (60.0%)	35 (66.0%)	n.s.
Male	20 (40.0%)	18 (34.0%)	n.s.
Symptom Severity (BCTQ)	0.6647 ± 0.0781	0.8792 ± 0.0735	<0.001
Functional Status (BCTQ)	15.92 ± 7.29	31.28 ± 6.47	<0.001

**Table 2 diagnostics-15-01417-t002:** Classification performance at optimal threshold (0.7).

Metric	Value
Accuracy	93.33%
Confusion Matrix	[[14, 1], [1, 14]]
Precision	0.93
Recall	0.93
F1-Score	0.93

**Table 3 diagnostics-15-01417-t003:** BCTQ Scores for normal and abnormal groups.

Group	Symptom Severity Scale (Mean ± SD)	Functional Status Scale (Mean ± SD)
Normal	0.6647 ± 0.0781 (0.5455–0.8182)	15.92 ± 7.29 (8–32)
Abnormal	0.8792 ± 0.0735 (0.7273–1.0000)	31.28 ± 6.47 (16–40)

**Table 4 diagnostics-15-01417-t004:** Distribution of SWMT values and CTS grades.

SWMT Threshold	n (%)	Interpretation
≤2.83	29 (29.6%)	Normal sensation
3.22–3.84	35 (35.7%)	Diminished light touch
4.08–4.56	24 (24.5%)	Diminished protective sensation
≥4.74	10 (10.2%)	Loss of protective sensation

**Table 5 diagnostics-15-01417-t005:** Demographic characteristics of participants in the magneto-electric field imaging study.

Characteristic	CTS Group (n = 32)
Age (years), mean ± SD	42.4 ± 14.9
Age Range (min–max)	18–77
Gender, n (%)	
Female	21 (65.6%)
Male	11 (34.4%)

**Table 6 diagnostics-15-01417-t006:** Features assessed to determine skin innervation.

Feature Category	Details
Significant Features	10 features (*p* < 0.05): *red_proportion*, yellow_green_proportion, avg_value, *haralick_2*, *haralick_3*, *haralick_5*, *haralick_6*, *haralick_7*, *haralick_9*, *haralick_12*
Borderline Feature	avg_grad_magnitude (*p* = 0.0503)
Non-Significant Features	9 features (*p* > 0.05): e.g., dark_proportion, edge_density, *haralick_4*

## Data Availability

Data supporting the reported results are available upon request due to privacy restrictions.

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
