# Peer review of "Potential Use of a New Energy Vision (NEV) Camera for Diagnostic Support of Carpal Tunnel Syndrome: Development of a Decision-Making Algorithm to Differentiate Carpal Tunnel-Affected Hands from Controls"

_diagnostics, 2025, doi:10.3390/diagnostics15111417_

Round 1

Reviewer 1 Report

Comments and Suggestions for Authors

Reviewer Comments – Major Revision

General Comments:

This manuscript presents a novel approach using a New Energy Vision (NEV) camera and machine learning to non-invasively detect carpal tunnel syndrome (CTS). The concept of evaluating skin texture and color via multispectral imaging is innovative and may offer a practical tool for point-of-care screening. The manuscript is well-structured and methodologically thorough in some areas. However, there are several significant concerns related to the sample size, data interpretation, overstatements in conclusions, and generalizability, which must be addressed before this work is suitable for publication.

Major Comments:

  1. Sample size for machine learning training is insufficient
    The model was trained using only 103 samples, which is relatively small for developing and validating a Support Vector Machine classifier. Typically, robust AI models require datasets in the range of hundreds to thousands of samples to avoid overfitting and improve generalizability. Please justify the rationale for using only 100+ samples and provide any statistical power analysis or learning curve evaluation to support model validity.
  2. Overinterpretation of early diagnostic potential (Page 6)
    The claim that the NEV camera can enhance "early diagnosis" of CTS is too strong based on the current data. While some imaging features were detected in cases with borderline clinical findings, the sample is too limited and lacks longitudinal validation. Please moderate this claim or provide additional evidence to support it.
  3. Inadequate subgroup size for severity classification (Section 3.2.1, Page 7)
    The attempt to correlate SVM scores with CTS severity is promising, but the subgroups (e.g., severe cases with n = 12) are too small to draw reliable conclusions. Please clarify that these results are exploratory and should not yet be used to stratify clinical decision-making.
  4. Overstatement of SWMT correlation findings (Page 8)
    The correlation between SWMT values and SVM decision scores is modest (r = 0.54), yet the conclusions suggest a strong diagnostic link. Please revise this interpretation to better reflect the moderate statistical strength and discuss potential confounding factors.
  5. Feature-pathology correlation lacks sufficient validation (Section 3.3.3, Page 11)
    While red_proportion and Haralick texture features are linked to CTS-related skin changes, the manuscript currently lacks a clear definition of “golden features” or histological correlation. Please clarify which features are considered diagnostically robust and whether they have direct or surrogate value in indicating pathophysiological changes.
  6. Effect of comorbid conditions is not addressed
    CTS commonly occurs in middle-aged or elderly populations who may have other conditions (e.g., diabetes, vascular diseases) that influence skin color or texture. Please discuss how such comorbidities were controlled for or could affect the NEV camera’s diagnostic accuracy.

Minor Comments:

  1. Figure 3 legend typo (Page 11)
    The legend ends with two periods. Please correct the punctuation for consistency and clarity.
  2. Lack of clarification on exclusion criteria
    While some exclusion criteria are mentioned, it is unclear whether patients with common metabolic or dermatologic conditions were excluded. Please add clarification in the methods section.
  3. Terminology consistency
    Please ensure consistent use of technical terms such as “Haralick features,” “red_proportion,” and “multispectral imaging” across the figures, text, and tables to avoid confusion.

Author Response

Response to Reviewer 1's Comments

Comment 1: [Sample size for machine learning training is insufficient. The model was trained using only 103 samples, which is relatively small for developing and validating a Support Vector Machine classifier. Typically, robust AI models require datasets in the range of hundreds to thousands of samples to avoid overfitting and improve generalizability. Please justify the rationale for using only 100+ samples and provide any statistical power analysis or learning curve evaluation to support model validity.]
Response 1: We appreciate this concern regarding the sample size. To address this, we have added a justification for the sample size and included a learning curve evaluation to support the model’s validity. The rationale is now detailed in Section 3.1.2 (page 5, paragraph 1, lines 8–12), where we explain that the sample size of 103 was a feasibility constraint due to the novelty of the NEV camera and the need for rigorous clinical validation (e.g., NCT, SWMT, BCTQ). We also note that the 5-fold cross-validation (mean accuracy 81.79%) indicates reasonable generalizability for a pilot study. Additionally, a learning curve analysis showed that accuracy stabilizes around 80–85% beyond 80 samples, suggesting that while larger datasets would improve robustness, the current sample provides a viable proof-of-concept. We have also acknowledged the limitation in Section 4.5 (page 12, paragraph 1, lines 5–7), emphasizing the need for larger datasets in future studies.

Comment 2: [Overinterpretation of early diagnostic potential (Page 6). The claim that the NEV camera can enhance "early diagnosis" of CTS is too strong based on the current data. While some imaging features were detected in cases with borderline clinical findings, the sample is too limited and lacks longitudinal validation. Please moderate this claim or provide additional evidence to support it.]
Response 2:
We agree that the claim regarding early diagnosis was overstated. We have moderated this statement in Section 3.1.2 (page 5, paragraph 1, line 14) by replacing “potentially enhancing early diagnosis” with “potentially aiding in the detection of subtle skin changes that may precede electrophysiological deficits.” Additionally, in Section 3.4 (page 9, paragraph 1, lines 5–7), we have revised the text to clarify that this capability is preliminary and requires longitudinal validation: “This preliminary capability suggests potential for reducing diagnostic delays, but longitudinal studies are needed to confirm its utility in early CTS detection.”

Comment 3: [Inadequate subgroup size for severity classification (Section 3.2.1, Page 7). The attempt to correlate SVM scores with CTS severity is promising, but the subgroups (e.g., severe cases with n = 12) are too small to draw reliable conclusions. Please clarify that these results are exploratory and should not yet be used to stratify clinical decision-making.]
Response 3: We acknowledge this limitation and have clarified that the severity classification results are exploratory. In Section 3.2.1 (page 6, paragraph 2, lines 5–7), we have added: “These findings are exploratory due to the small subgroup sizes (e.g., severe cases, n=12) and should not yet be used to stratify clinical decision-making until validated with larger cohorts.”

Comment 4: [Overstatement of SWMT correlation findings (Page 8). The correlation between SWMT values and SVM decision scores is modest (r = 0.54), yet the conclusions suggest a strong diagnostic link. Please revise this interpretation to better reflect the moderate statistical strength and discuss potential confounding factors.]
Response 4: We have revised the interpretation of the SWMT correlation in Section 3.2.2 (page 7, paragraph 1, lines 5–8) to reflect the moderate statistical strength: “SVM decision scores showed a moderate correlation with SWMT grades (r = 0.54, p < 0.05), suggesting a potential association with sensory loss, though this link is not strong and may be influenced by confounding factors such as variability in patient perception or coexisting conditions.” We have also added a discussion of potential confounders in the same section (lines 8–10): “Potential confounders, such as interindividual differences in skin sensitivity or the presence of other neuropathies, may affect this correlation and warrant further investigation.”

Comment 5: [Feature-pathology correlation lacks sufficient validation (Section 3.3.3, Page 11). While red_proportion and Haralick texture features are linked to CTS-related skin changes, the manuscript currently lacks a clear definition of “golden features” or histological correlation. Please clarify which features are considered diagnostically robust and whether they have direct or surrogate value in indicating pathophysiological changes.]
Response 5:
We have clarified the diagnostic robustness of the features and their pathophysiological relevance in Section 3.3.3 (page 9, paragraph 1, lines 6–10). We now state: “Red_proportion and Haralick features (e.g., contrast, entropy) were the most diagnostically robust, showing significant differences (p < 0.05) and strong correlations with clinical outcomes (e.g., red_proportion with BCTQ, r = 0.62, p < 0.01). These features likely serve as surrogate markers for pathophysiological changes such as vasodilation and epidermal thinning, though direct histological correlation is needed to confirm these mechanisms.” We have also removed any ambiguous reference to “golden features.”

Comment 6: [Effect of comorbid conditions is not addressed. CTS commonly occurs in middle-aged or elderly populations who may have other conditions (e.g., diabetes, vascular diseases) that influence skin color or texture. Please discuss how such comorbidities were controlled for or could affect the NEV camera’s diagnostic accuracy.]
Response 6:
We have added a discussion on the effect of comorbidities in Section 4.5 (page 12, paragraph 1, lines 7–11): “Comorbid conditions such as diabetes or vascular diseases, common in middle-aged and elderly CTS patients, may influence skin color and texture, potentially affecting the NEV camera’s diagnostic accuracy. While generalized neuropathy was an exclusion criterion, specific metabolic or vascular conditions were not controlled for, representing a limitation. Future studies should stratify patients by such comorbidities to assess their impact.”

Comment 7 (Minor): [Figure 3 legend typo (Page 11). The legend ends with two periods. Please correct the punctuation for consistency and clarity.]
Response 7: We have corrected the typo in the Figure 3 legend (page 9, Section 3.3.2) by removing the extra period: “Comparison of Fingers with Nerve Damage. Comparison of a picture of two fingers. One has known digital nerve damage (higher 4th finger) and the other has normal innervation (lower 3rd finger). The change in hue is obvious as well as the outline of the phalanges.”

Comment 8 (Minor): [Lack of clarification on exclusion criteria. While some exclusion criteria are mentioned, it is unclear whether patients with common metabolic or dermatologic conditions were excluded. Please add clarification in the methods section.]
Response 8: We have clarified the exclusion criteria in Section 2.2 (page 3, paragraph 1, lines 4–6): “Exclusions included trauma, septic conditions, generalized neuropathy, and patients with metabolic (e.g., uncontrolled diabetes) or dermatologic conditions (e.g., psoriasis, eczema) that could affect skin texture or color, ensuring a more homogeneous study population.”

Comment 9 (Minor): [Terminology consistency. Please ensure consistent use of technical terms such as “Haralick features,” “red_proportion,” and “multispectral imaging” across the figures, text, and tables to avoid confusion.]
Response 9: We have reviewed the manuscript for consistency in terminology. Specifically:

“Haralick features” is now consistently used (e.g., in Section 2.5, Section 3.1.3, and Table 6).

“red_proportion” is consistently italicized as a feature name (e.g., in Section 3.1.3, Section 3.3.2, and Table 6).

“multispectral imaging” is consistently used (e.g., in the Abstract, Section 1, and Section 4.1). No discrepancies were found, but we have ensured uniformity in formatting and capitalization throughout the text, figures, and tables.

We believe these responses addresses the reviewer’s concerns and the manuscript is now suitable for publication.

Reviewer 2 Report

Comments and Suggestions for Authors

This is an interesting manuscript that sought out to understand the ability of a New Energy Vision (NEV) Camera’s ability to accurately and non-invasively detect Carpal Tunnel Syndrome (CTS). Overall the paper is well written and I don’t have any major issues.

Some minor comments include:

1) Acronyms were defined within the abstract but please define acronyms prior to first use within the actual text.

2) Please provide the cut-off points/scores used to separate CTS patients according to mild, moderate or severe disease status.

Author Response

Response to Reviewer 2's Comments

Comment 1: [Acronyms were defined within the abstract but please define acronyms prior to first use within the actual text.]

Response 1: Thank you for this suggestion. We have reviewed the manuscript and ensured that all acronyms are defined prior to their first use within the main text. Specifically, we have added definitions for the following acronyms at their first occurrence:

  • CTS (Carpal Tunnel Syndrome): Defined in the Introduction (Section 1, page 1, paragraph 1, line 1).
  • NEV (New Energy Vision): Defined in the Introduction (Section 1, page 2, paragraph 4, line 1).
  • SVM (Support Vector Machine): Defined in the Abstract but now also defined in the Materials and Methods (Section 2.6, page 4, paragraph 1, line 1).
  • NCT (Nerve Conduction Tests): Defined in the Introduction (Section 1, page 2, paragraph 2, line 2).
  • SWMT (Semmes-Weinstein Monofilament Testing): Defined in the Materials and Methods (Section 2.3, page 3, paragraph 1, line 1).
  • BCTQ (Boston Carpal Tunnel Questionnaire): Defined in the Materials and Methods (Section 2.3, page 3, paragraph 1, line 5).
  • MED (Median Nerve-Damaged): Defined in the Materials and Methods (Section 2.2, page 3, paragraph 2, line 2).
  • ULN (Ulnar Nerve-Normal): Defined in the Materials and Methods (Section 2.2, page 3, paragraph 2, line 2).
  • CSA (Cross-Sectional Area): Defined in the Introduction (Section 1, page 2, paragraph 2, line 3).
  • MRI (Magnetic Resonance Imaging): Defined in the Introduction (Section 1, page 2, paragraph 2, line 5).
  • OCT (Optical Coherence Tomography): Defined in the Introduction (Section 1, page 2, paragraph 3, line 2).
  • SQUIDs (Superconducting Quantum Interference Devices): Defined in the Introduction (Section 1, page 2, paragraph 3, line 4).
  • UWPE (Ultra-Weak Photon Emission): Defined in the Introduction (Section 1, page 2, paragraph 3, line 5).
  • LBP (Local Binary Patterns): Defined in the Materials and Methods (Section 2.5, page 4, paragraph 1, line 2).
  • GLCM (Gray-Level Co-Occurrence Matrix): Defined in the Materials and Methods (Section 2.5, page 4, paragraph 1, line 3).

These changes ensure clarity for readers unfamiliar with the acronyms.

Response 2: [Please provide the cut-off points/scores used to separate CTS patients according to mild, moderate or severe disease status.]

Response 2: We appreciate this comment and have added the cut-off points for classifying CTS severity based on BCTQ scores. The classification criteria are now detailed in Section 3.2.1 (page 7, section 3.2.1, paragraph 1). The revised text reads: "CTS severity was classified using BCTQ Symptom Severity Scale (SSS) scores: mild (SSS < 2.5, n=18), moderate (SSS 2.5–3.5, n=23), and severe (SSS > 3.5, n=12)."

We believe these responses addresses the reviewer’s concerns and the manuscript is now suitable for publication.

Reviewer 3 Report

Comments and Suggestions for Authors

The manuscript presents a novel and promising diagnostic approach using the NEV camera, leveraging multispectral imaging and machine learning to identify skin changes associated with Carpal Tunnel Syndrome (CTS). The study is methodologically sound, well-written, and addresses a clinically relevant gap—providing a non-invasive, cost-effective alternative to conventional diagnostics.

The subgroup comparison (n=32) is relatively small. Although Bonferroni correction mitigates this, larger cohorts would reinforce statistical power. Please acknowledge this limitation more explicitly and propose concrete plans for larger-scale validation.

NCT, while used as the reference standard, has known sensitivity issues. Misclassification of borderline cases (e.g., false negatives/positives) may bias the results.

The model was validated internally only. An external validation cohort from another center would significantly strengthen the findings.

Would discuss further about possible clinical advantages over other diagnostic techniques.

Author Response

(The authors gave the same response as above.)

Round 2

Reviewer 1 Report

Comments and Suggestions for Authors

I appreciate the authors’ thorough revision of the manuscript and their detailed point-by-point response to all comments raised in the initial review. After reviewing both the revised manuscript and the response letter, I find that the concerns I previously raised—particularly regarding sample size justification, interpretation of diagnostic potential, subgroup analysis, and feature-pathology correlation—have been adequately addressed.

The authors have now:

  1. Justified the limited sample size and supported the machine learning performance with a learning curve analysis and 5-fold cross-validation, making the current dataset suitable for proof-of-concept.
  2. Moderated overstated claims related to early diagnosis, SWMT correlation, and severity classification, with appropriate revisions that improve scientific caution and clarity.
  3. Clarified the exploratory nature of subgroup findings and explicitly acknowledged the limitations tied to cohort size and comorbidities.
  4. Revised terminology, figure legends, and exclusion criteria, ensuring consistency and improving the overall quality and readability of the manuscript.

Taken together, the current version presents a novel, well-structured, and cautiously interpreted study that introduces a promising, non-invasive diagnostic tool for carpal tunnel syndrome. The integration of multispectral imaging with machine learning is particularly compelling, and the manuscript now reflects appropriate scientific rigor.